# Effect of Cell Therapy and Exercise Training in a Stroke Model, Considering the Cell Track by Molecular Image and Behavioral Analysis

**DOI:** 10.3390/cells11030485

**Published:** 2022-01-30

**Authors:** Mariana P. Nucci, Fernando A. Oliveira, João M. Ferreira, Yolanda O. Pinto, Arielly H. Alves, Javier B. Mamani, Leopoldo P. Nucci, Nicole M. E. Valle, Lionel F. Gamarra

**Affiliations:** 1Hospital Israelita Albert Einstein, São Paulo 05652-000, Brazil; mariana.nucci@hc.fm.usp.br (M.P.N.); fernando.anselmo@einstein.br (F.A.O.); joaomatiasferreirav@gmail.com (J.M.F.); yoliveira1160@gmail.com (Y.O.P.); ariellydahora1997@gmail.com (A.H.A.); javierbm@einstein.br (J.B.M.); nicolemev@gmail.com (N.M.E.V.); 2LIM44, Hospital das Clínicas da Faculdade Medicina da Universidade de São Paulo, São Paulo 05403-000, Brazil; 3Centro Universitário do Planalto Central, Brasília 72445-020, Brazil; leopoldo.nucci@gmail.com

**Keywords:** cell therapy, physical exercise, multimodal nanoparticles, stroke, behavioral test, human bone marrow mesenchymal stem cells, near-infrared fluorescence image, bioluminescence

## Abstract

The goal of this study is to see how combining physical activity with cell treatment impacts functional recovery in a stroke model. Molecular imaging and multimodal nanoparticles assisted in cell tracking and longitudinal monitoring (MNP). The viability of mesenchymal stem cell (MSC) was determined using a 3-[4,5-dimethylthiazol-2-yl]-2,5 diphenyl tetrazolium bromide (MTT) assay and bioluminescent image (BLI) after lentiviral transduction and MNP labeling. At random, the animals were divided into 5 groups (control-G1, and experimental G2-G5). The photothrombotic stroke induction was confirmed by local blood perfusion reduction and Triphenyltetrazolium chloride (TTC), and MSC in the G3 and G5 groups were implanted after 24 h, with BLI and near-infrared fluorescence image (NIRF) tracking these cells at 28 h, 2, 7, 14, and 28 days. During a 28-day period, the G5 also conducted physical training, whereas the G4 simply did the training. At 0, 7, 14, and 28 days, the animals were functionally tested using a cylinder test and a spontaneous motor activity test. MNP internalization in MSC was confirmed using brightfield and fluorescence microscopy. In relation to G1 group, only 3% of cell viability reduced. The G2–G5 groups showed more than 69% of blood perfusion reduction. The G5 group performed better over time, with a progressive recovery of symmetry and an increase of fast vertical movements. Up to 7 days, BLI and NIRF followed MSC at the damaged site, demonstrating a signal rise that could be connected to cell proliferation at the injury site during the acute phase of stroke. Local MSC therapy mixed with physical activity resulted in better results in alleviating motor dysfunction, particularly during the acute period. When it comes to neurorehabilitation, this alternative therapy could be a suitable fit.

## 1. Introduction

Ischemic stroke still represents a serious threat to human health and economic systems, with an estimated 116 million healthy lives lost each year due to stroke-related death and disability [1]. According to the Global Burden of Disease (GBD), Brazil had 23.9 incidences of stroke per 100,000 people in 2007 [2]. Stroke is one of the three leading causes of death and disability among age-related non-communicable diseases worldwide, followed by ischemic heart disease, and dementia [3]. Treatment for functional rehabilitation after stroke is still limited, despite breakthroughs in pharmacological and surgical therapy [4]. Early reperfusion with thrombolytic or endovascular thrombectomy which can reduce penumbra injury and improve clinical outcomes, remains the gold-standard therapy for stroke. Only a few patients receive effective treatment due to the limited therapeutic time window (8 h). In this scenario, novel treatments aimed at promoting brain balance, reducing the infarct region, and accelerating neurological recovery appear to be urgently needed [5,6].

Therapy based on mesenchymal stem cells (MSC) has emerged as a new treatment strategy for stroke due to their unique properties that include easy isolation, low immunogenicity, good proliferation, multipotent differentiation potential, and strong paracrine capacity [4]. MSC positive benefits are related to paracrine and immunomodulatory processes, which have been widely recognized and investigated [7] as a valuable therapeutic strategy in instances where a pharmacological window is no longer possible.

MSC have been shown in preclinical studies to be unable to replace dead neurons after ischemic events; however, they provide a variety of other benefits over parallel processes, including up-regulation of growth factor at the injured site, decreased apoptosis, reduced glial scar formation, promoted axonal outgrowth, synaptic remodeling, neurogenesis, angiogenesis, and astrocyte and oligodendrocyte growth factors [8], as well as improved brain function, through neuronal ischemia protection and secondary brain damage [4].

Physical exercise is another treatment option that is commonly used during stroke recovery. It is important in the recovery from a variety of deficits, such as motor dysfunction, which is mediated by neural plasticity [9]. This approach not only improves muscular and cardiovascular health, but it also has the potential to improve neuroplasticity and decrease cognitive decline in the brain [10]. The primary neuroprotective mechanisms linked to exercise may play a role in the acute phase [11], boosting the amount of MSC circulating and mobilizing them, as well as having favorable effects on the proliferation of developmentally early stem cells [12]. The physiological stress induced by acute and high-intensity exercise dramatically increases MSC and promotes MSC mobilization from bone marrow to peripheral circulation until 15 min after the exercise ceases [12].

It should be noted that physical activity also enhances MSC therapeutic action, by inhibiting apoptosis of cells, including neuronal cells and transplanted MSC themselves, improving neurological function [12], as well as MSC tracking into the brain lesion region, because the efficacy of cellular therapy is dependent on its homing ability and engraftment into the injury site. Generally, studies about cell homing/tracking after ischemia injury have used ex vivo techniques to quantify MSC in recipient animals; however, this form of analysis cannot yield longitudinal data about the migration or graft dynamic in a single animal. Noninvasive molecular imaging techniques, such as magnetic resonance imaging (MRI), positron emission tomography (PET), single-photon emission computed tomography (SPECT), near-infrared fluorescence (NIRF) imaging, and bioluminescence imaging (BLI) could be used to home and track these cells in vivo by labeling cells with a contrast agent [13]. 

In preclinical studies, cell labeling with superparamagnetic iron oxide nanoparticles (SPION) has been widely used as a tracking method, and its combination with near-infrared (NIR) dye, known as a bimodal technique, demonstrated a greater ability of penetration and revealed deeper tissues [14]. With the use of innovative NIRF probes and optical instruments, in vivo fluorescence imaging has grown significantly, allowing for the evaluation of dynamic migration and distribution of transplanted MSC as well as tissue regeneration based on stem cells [13]. Bimodal analysis may necessitate genetic cell modification to express luciferase enzyme signal [15], allowing BLI technique to be used as a supplement to NIRF imaging, therefore improving ischemic and inflammatory monitoring, as well as tracking viable cells after engraftment reduction [13]. 

Behavioral testing, which has provided useful information about the biological basis and prospective rehabilitation efforts, as well as being essential in the development of translational therapies, is another widely used tool to validate damage and recovery in stroke models [16].

Therefore, the purpose of this study is to analyze the effect of physical activity associated with cell therapy on the homing and tracking of cells, as well as functional recovery, in a stroke model.

## 2. Materials and Methods

### 2.1. Isolation, Culture and Immunophenotypic Characterization of Human Bone Marrow Mesenchymal Stem Cells

The MSC extraction and use were approved by the Ethics Research Committee of Instituto Israelita de Ensino e Pesquisa Albert Einstein (Sao Paulo, Brazil) under CAAE number: 64288917.0.0000.0071. The MSC isolation, culture and immunophenotypic characterization were performed in previous study [17]. In brief, mononuclear cells were obtained from bone marrow of healthy donors and separated by density gradient using Ficoll-Paque™ Premium (1.084 g/mL) (GE Healthcare, Uppsala, Sweden). After isolation, mononuclear cells were cultivated in Dulbecco’s Modified Eagle’s Medium/Nutrient Mixture F-12 (DMEM/F-12) (GIBCO^®^ Invitrogen Corporation, Carlsbad, CA, USA) supplemented with fetal bovine serum (FBS) (GIBCO^®^ Invitrogen Corporation, CA, USA), and 1% antibiotic-antimycotic solution (GIBCO^®^ Invitrogen Corporation, CA, USA), incubated at 37 °C with 5% de CO_2_. When required, the cells were trypsinized using TrypLE™ Express Enzyme (GIBCO^®^ Invitrogen Corporation, CA, USA). The immunophenotypic characterization was performed after third cell passage using flow cytometry (FACSAria™ III, BD Biosciences, San Jose, CA, USA), and data were analyzed through FlowJo software (Tree Star, Ashland, OR, USA).

### 2.2. Lentiviral Transduction of MSC

MSC were genetically modified to express luciferase using a vesicular stomatitis virus G glycoprotein (VSV-G) responsible to carry the lentiviral vector pMSCV_Luc2_T2A_Puro. This vector obtained from pseudotyped viruses encodes the bioluminescent reporter, luciferase-2, and the puromycin gene N-acetyl-transferase, which confers resistance to puromycin. The production of virions used in lentiviral transduction was done as previously described [18]. For lentiviral transduction, a 24-well plate was used, and 2 × 10^5^ MSC distributed per well were kept in DMEN/F-12 medium with 10% SFB until achieving 70% of confluency. Then, the fresh culture medium with lentiviral vector at a multiplicity of infection of 10 (MOI = 10) in the presence of 8 µg per mL of polybrene (Sigma–Aldrich, St. Louis, MO, USA) was added to each well. The plate remained in culture overnight and the medium was exchanged the day after, looking to remove viral particles. After 48 h, transduced MSC selection was initiated with 1 μg per mL of puromycin addition every two days for one week. After that, only MSC expressing luciferase protein (MSC_Luc_) and puromycin resistance remained in culture.

### 2.3. Nanoparticles Used in the MSC Homing Evaluation

The MSC were labeled with magnetic nanoparticles (MNP) composed by a crystalline iron oxide (Fe_3_O_4_) nucleus of 8 nm coated with dextran and presenting an average hydrodynamic size of 35 nm and a zeta potential of +31 mV. The MNP are conjugated with two fluorophores that emit excitation/emission wavelengths in the 750/777 nm range (NIR spectrum) and in the 558/580 nm range (visible spectrum) (Biopal, Molday ION™, Worcester, MA, USA). Nanoparticle characteristics such as hydrodynamic size, zeta potential, and optical properties were approached in a previous study [19].

### 2.4. Cell Labeling and the Internalization Evaluation of MSC Labeled with MNPs

For MSC labeling with MNP, MSC were cultivated in a 24-well plate until achieving 70% confluency. Then, the plate was washed with phosphate-buffered saline (PBS) (GIBCO^®^ Invitrogen Corporation, CA, USA), and fresh medium with 50 µg Fe/mL of MNP was added. After 18 h of incubation, the plate was washed 3 times with PBS to remove the MNP not internalized. 

MNP internalization was evaluated by brightfield and fluorescence microscopy, and NIRF imaging. For brightfield microscopy analysis, the MSC labeled were fixed through incubation in PBS (Sigma-Aldrich, St Louis, MO, USA) containing 4% paraformaldehyde, then, 500 µL Prussian blue staining solution containing 0.25 mg potassium ferrocyanide [K_4_Fe(CN)_6_] (Sigma–Aldrich, St Louis, MO, USA) and 5% hydrochloric acid (Merck, Darmstadt, Germany) was added for 5 min, followed by washing with PBS. Prussian blue staining highlights the internalized nanoparticles in the cells by coloring iron with blue. For fluorescence microscopy evaluation, 4,6-diamidino-2-phenylindole dihydrochloride (DAPI) was added on plate for 5 min, also followed by PBS washing. After that, images were acquired using an excitation/emission filter of 358/461 nm for DAPI and another of 530/550 nm for MNP. The brightfield and fluorescence images were obtained under Nikon TiE microscopy (Nikon, Tokyo, Japan). Finally, for NIRF internalization evaluation, MSC were trypsinized and transferred to a black 96-well plate that was positioned in the IVIS^®^ Lumina LT Series III equipment (Xenogen Corp., Alameda, CA, USA). NIRF images were acquired using automatic exposure time, F/stop 4, binning 8, field of view (FOV) of 12.9 cm, excitation filter of 710 nm and emission of 780 nm. The NIRF intensity (photons/s) was selected from a region of interest (ROI) of 2.5 cm^2^. 

### 2.5. Cellular Viability of MSC Labeled with MNP

The cellular viability of MSC labeled with MNP was performed in quadruplicate though MTT (3-[4,5-dimethylthiazol-2-yl]-2,5 diphenyl tetrazolium bromide) and BLI assays. MTT assay was performed using the protocol previously described by Lindemann et al. [20] The percentage of cell viability was obtained from the mean absorbance of MSC labeled in relation to the mean absorbance of MSC unlabeled controls ((labeled sample/control) × 100).

For cell viability evaluation by BLI, 20 µL of d-luciferina (150 mg/mL) (XenoLight, Perkin Elmer, Boston, MA, USA) was added in each well and BLI images acquired in IVIS^®^ Lumina LT Series III equipment with the same parameters described in item 2.4. The cellular viability was obtained by the relationship: (BLI intensity of MSC labeled/BLI intensity of MSC unlabeled) × 100.

### 2.6. Animals and Experimental Design

We used male Wistar rats that were 2 months old and weighed roughly 250–300 g. The animals were kept at the Centro de Experimentação e Treinamento em Cirurgia (CETEC) vivarium, which is part of the Instituto Israelita de Ensino e Pesquisa Albert Einstein (IIEP) and were exposed to 21 ± 2 °C room temperature, 60 ± 5% relative humidity in full ventilation, a 12 h light/dark cycle (7 a.m.–7 p.m.), and with access to food and water ad libitum. The Association for Assessment and Accreditation of Laboratory Animal Care International (AAALAC International) has accredited the vivarium, and the general conditions were evaluated daily. The study was approved by the Ethics in Animal Research Committee of the Hospital Israelita Albert Einstein (Sao Paulo, Brazil), number 2963-17. 

As indicated in Figure 1, the experimental design performed two temporally separate evaluations, one before and one after stroke induction. Control behavioral tests were performed in all animals, randomly allocated, coded, and housed in individual cages, prior to stroke induction, with the goal of selecting good runners for the experimental groups (G4 and G5), condition required to perform the physical training following stroke induction. A control brain perfusion was performed immediately before surgery to assess pre-ischemic induction in the stroke experimental groups (G2–G5).

Animal (G2–G5) blood–brain perfusion was reanalyzed as soon as stroke induction was completed. TCC (2,3,5-triphenyltetrazolium chloride) staining was used to assess focal brain ischemic lesion in part of animals one day later, and MSC were administrated in the G3 and G5 groups. These same groups had molecular image analysis using bioluminescence and near-infrared fluorescence 4 h, as well as at 1, 3, 5, 7, 14, and 21 days after MSC administration. The treadmill physical training was performed 5 days a week for 4 weeks in the G4 and G5 groups, and behavioral tests were done at 7, 14, 21, and 28 days after stroke induction in all groups (G1–G5).

### 2.7. Treadmill Habituation Evaluation (Selection of the Best Running Animals for the Experiment)

The animals were habituated to the treadmill three days prior to stroke induction, during the same time of the day, and after 30 min of acclimatization in the behavioral room. The rodent motorized treadmill equipment (AVSProjetos, Sao Paulo, SP, Brazil) features six lanes, each with front- and back-wall air opening to allow for uncontrolled airflow. Speed adjustments affected the six lanes of the treadmill belt, with six animals evaluated per time, with 0 degrees of lane slope, without the electrical shock. After 1 min of environment exploration, the habituation was performed for 5 min. Furthermore, rats were urged to run by giving them a light push around the tail, especially in the early days of becoming used to jogging on the treadmill. The Dischman scale (score from 1 to 5) [21] and the position of the rat’s tail (score 0.25 to 1) [22] were used to evaluated the animal’s performance at the end of the test as an indicator of tiredness based on the effort. The best values obtained for three days of the evaluation were used to identify the best runners after adding all parameter data.

### 2.8. Induction of Stroke through Photothrombosis

A photothrombotic model was used to generate focal ischemic injury, which required the injection of a photosensitive dye (Rose Bengal) intraperitoneally into bloodstream and the application of a cold light source to the brain area of interest (pre-central gyrus). As a result, the photosensitive dye becomes activated, causing endothelium damage as well as platelet activity and thrombosis, resulting in local blood flow disruption. The animals were anesthetized with isoflurane gas (4% for induction and 1.8% during surgery), placed in a stereotaxic apparatus (Harvard Apparatus, Holliston, MA, USA), shaved on the surgical site, and then a craniectomy with an electric drill was performed, exposing a bone window in the left somatosensory cortex (+2 to −2 mm in the anterior–posterior direction and +2 mm on the medial–lateral axis from the Bregma, according to the atlas of Paxinos and Watson) [23]. Rose Bengal (50 mg/mL) (Sigma–Aldrich) was supplied according to earlier calculations (V = WW × 0.1 where: V: volume to be administered; W is the weight of the animal measured before the procedure), and 20 min before the initiation of laser application, time required for the photosensitive dye to penetrate the blood–brain barrier. The laser (light source, Hamamatsu) was positioned 4–5 mm from the dura mater, under the region of interest and used for 65 min with the following parameters: 50 mm output, and 50% laser power. Local blood perfusion was re-evaluated at the conclusion of induction, the surgical incision was sutured, and the tramadol analgesic (5 mg/kg) was administrated (i.v.). The animals were placed on a heating pad throughout anesthesia to keep their rectal temperature at 37.0 ± 0.5 °C (PhysioSuite, Kent Scientific Corporation, Torrington, CT, USA).

### 2.9. Focal Ischemic Lesion Induction Evaluation by Blood Perfusion and TTC Staining

Local blood perfusion analysis with a PeriCam Perfusion Speckle Imager (PSI) system (Perimed, Stockholm, Sweden) was used to confirm the focal ischemic brain lesion, comparing baseline perfusion to perfusion in the ischemic areas after induction (effective ischemia being considered to be around 70% of blood decrease) [24,25]. As described in a previous study [26], TTC staining was conducted after 24 h of ischemic lesion induction. The infarcted area is indicated by TTC-unstained brain tissue, while the viable brain area is indicated by TTC-stained tissue [27].

### 2.10. MSC Administration

After 24 h of focal stroke induction, animals were anesthetized with isoflurane and subjected to cell implantation (1 × 10^6^ cells disperse in 10 µL of DMEM-F12). The surgical incision was reopened and the MSC were slowly administered around the ischemic lesion with a Hamilton syringe (10 µL per 3 min), finishing the procedure with the incision suture.

### 2.11. Treadmill Protocol—Exercise Training

All animals in the experimental group (G4 and G5) or in the control group (G1–G3) had a 28-day period with or without treadmill physical exercise, respectively, after focal ischemia brain induction. Animals in the control group were kept in standard cages, while those in the experimental group were made to run for 15 min with a 1 min warm-up. The maximum speed of each animal in the G4 and G5 groups was tested on the first day of each week of training, and the parameter used to adjust the speed training of each week established a progressive charge increase of training over weeks with the using 50%, 60%, 70% and 80% of the maximum speed evaluated in each week. The Dischman scale and the position of the rat’s tail were used to assess animal performance during training.

### 2.12. Behavioral Analysis

The spontaneous locomotor activity and cylinder tests were included in behavioral analysis. Both were carried out in the assessment room after 30 min of animal acclimatization, in the dark and at the same time of day, between 4 p.m. and 5 p.m. The configuration and parameter acquisition reported in a previous study [18] were used to quantify spontaneous global locomotor activity using Infrared (IR) Actimeter LE 8825 devices (Actitrack, Panlab Harvard Apparatus, Barcelona, Spain). Slow movements (S-MOV), fast movements (F-MOV), slow stereotyped (S-STE), fast stereotyped (F-STE) slow rearing (S-REA), and fast rearing (F-REA) were the six parameters used to compare groups and sessions. The data were processed using SEDACOM software v2.0. (Panlab Hardvard Apparatus, Barcelona, Spain). 

The cylinder test was used to assess forelimb asymmetries. All movements of rats were recorded for 5 min in a transparent glass cylinder ~20 cm in diameter and 30 cm high). The animal’s movements were watched from all sides with an angled mirror placed behind the cylinder. The recordings were subjected to a blind analysis, with each time a forelimb was placed on the cylinder wall being counted, and only the first 20 trials being considered. As described by Schallert [28], performance was scored as the probability of reduced forelimb usage: (ipsi − contra)/(ipsi + contra + both) × 100.(1)

### 2.13. MSC Homing after Implantation by Bioluminescence and Fluorescence Images

After 4 h, 1, 3, 5, 7, 14, and 21 days of cell implantation, MSC labeled with MNP followed using BLI and NIRF. IVIS equipment was used to capture both image modalities. D-luciferin (150 mg/kg) was delivered through the animal’s intraperitoneal site 10 min before acquisition of BLI images. They were then anesthetized with 5% of isoflurane, positioned in the equipment, and kept 2% anesthetized during image acquisition. The following parameters were used to acquire the BLI and NIRF images: automated exposure time, F/stop 4, binning 8 and FOV 12.9 cm.

### 2.14. Statistical Analysis

In each analysis, the data were reported as the mean and standard deviation. To determine variance equality, Levene’s tests were used. Behavioral analysis across groups was compared using the ANOVA test, followed by Bonferroni-corrected post hoc testing, and all analysis was considered at the 0.05 level of significance. JASP software v0.14.1 (http://www.jasp-stats.org, accessed in November 2021) was used for all statistical analysis.

## 3. Results

### 3.1. Evaluation of MNP Internalization into MSC and Their Viability

The evaluation of MNP internalization into MSC allowed the verification of nanoparticle magnetic and fluorescent characteristics, as shown in Figure 2. In the MSC unlabeled with MNP images (Figure 2A,B), it is possible to visualize the cell nucleus in blue (DAPI) by fluorescence microscopy (Figure 2A) and the absence of Prussian blue staining (Figure 2B) due to MNP absence. In images of MSC labeled with MNP (Figure 2C,D), fluorescence microscopy allowed the visualization of MNP internalized into MSC cytoplasm in red due to the presence of fluorophore in the visible spectrum (558/580 nm) using the TRITC filter, and the MSC nucleus highlighted in blue by DAPI staining (Figure 2C). Moreover, the brightfield microscopy image made it possible to corroborate the findings of fluorescence microscopy, demonstrating the internalized MNPs in MSC cytoplasm stained in blue (Figure 2D). MNP internalization into MSC was also confirmed by NIRF, which obtained a value in the order of 10^8^ photons/s for MSC labeled (Figure 2E).

MSC viability after labeling with 50 µg Fe per mL of MNP was analyzed by MTT and BLI assays. The graphic of Figure 2F and Table 1 shows 97.6% of MSC labeled viability compared to control by MTT; on the other hand, MSC labeled viability accessed by BLI was 96.9% in relation to control (MSC unlabeled).

### 3.2. Evaluation of Focal Ischemic Brain Lesion Induction by Phototrombosis, Using Local Blood Perfusion Analysis and TTC Staining

Focal ischemic brain lesion was acutely assessed by decreasing local blood flow, and comparing blood perfusion of the region of interest before (Figure 3A,C,E) and after ischemic induction (Figure 3B,D,F). All stroke groups G2–G5 showed a decrease in local blood perfusion after induction of greater than 69%, as shown in the graphic of Figure 3G and values of Table 2, with no statistical difference between groups regarding the decrease in in local blood perfusion after induction (*p* = 0.742; Table 3). In the two-by-two comparison using the post hoc test, no differences were observed between groups. Equal variance tests (Levene’s) were used to assess the differences in blood perfusion levels between groups (*p* = 0.989).

### 3.3. Performance in Treadmill Training

Physical training was centered on increasing the difficulty of physical activity in accordance with the animal’s weekly performance. The maximum speed (Table 4) was measured on the first day of each week’s training and compared over time and between groups. As demonstrated in Figure 4 and Table 5, there was a significant difference in time factor (weeks: *p* < 0.001) and group (G4 and G5: *p* = 0.007), but no difference in the group and week interaction (*p* = 0.295) by ANOVA test.

### 3.4. Behavioral Evaluation by the Cylinder Test and the Spontaneous Locomotor Activity Test

Using the symmetric score, the cylinder test showed that there was no significant difference between groups in basal time by ANOVA test (*p* = 0.489), but there was a significant difference in posterior timings (at 7, 14, 21, and 28 days) (*p* < 0.001). The primary differences were found between the experimental groups (G2–G5) with the sham group (G1) (*p* < 0.001), as well as the “stroke” group versus the “stroke + MSC” group (G2 vs. G3: at 7 days: *p* = 0.002; at 14-day: *p* = 0.001; at 21 days: *p* = 0.030; at 28 days: *p* < 0.001), and versus the “stroke + MSC + exercise group” (G2 vs. G4: at 7, 14, 21, and 28 days: *p* < 0.001) by the post hoc tests. As demonstrated in Figure 5A, there was a significant difference in symmetric scores between the “stroke + exercise” and “stroke + MSC + exercise” groups at 7, 14, 21 days (G4 vs. G5: *p* = 0.004), and at 28 days (G4 vs. G5: *p* < 0.001). However, there was no significant difference between the “stroke” and “stroke + exercise” groups (G2 vs. G3), nor between the latter and the “stroke + MSC” group (G3 vs. G4).

As indicated in Figure 5B–E, the spontaneous locomotor activity analysis was based on the frequency of horizontal and vertical animal movements (slow and fast), comparing groups and assessment times. All the baseline measures showed that the groups performed similarly. In terms of horizontal movements, the frequency of both slow and fast movements reduced over time in all groups, most likely due to animal habituation with the test. Both the “stroke + MSC + exercise” (green circles) and “sham” (black squares) groups, in contrast, showed a greater reduction in slow movements (green circles and black squares in Figure 5B). On the other hand, a subtle decrease in fast movements (green circles and black squares in Figure 5C) was observed, implying that animals who received cells and exercised had a better recovery in horizontal exploring agility than those who only received cells (blue squares in Figure 5B,C) or exercise (pink triangles in Figure 5B,C), exhibiting a pattern similar to the stroke group (red stars in Figure 5B,C). Despite this, the group that only received cells (blue squares in Figure 5B,C) performed somewhat better than the group that simply exercised (pink triangles in Figure 5B,C). 

The frequency of slow (Figure 5D) and fast (Figure 5E) vertical movements did not vary significantly over time, according to visual examination (Figure 5D,E). Slow movements continued to discretely decrease after a substantial decline in movement frequency at the start of the analysis due to stroke induction (at day 7 of Figure 5D,E). The fast movements of the experimental groups (all symbols except the black squares of Figure 5E) had a great recovery with frequency increase at 14 days, though the group that received cells and exercise (green circles in Figure 5E) had a higher recovery, approaching the movement pattern of the sham group (back squares in Figure 5E), compared to the other experimental groups, which only received cells (blue squares in Figure 5E) or exercise only (red stars in Figure 5E). As a result, slow movements continued to deteriorate over time following the initial decline, probably as a compensation for the rise in fast movements. Only fast vertical movement had a continual increase in frequency after the 7th day of evaluation, indicating that the acute-stage recovery was more important than the test habituation impact.

### 3.5. Noninvasive Evaluation of Homing of MSC_Luc_ Labeled with MNP by Molecular Images in the Stroke Model

The tracking of MSC labeled with MNP was performed weekly until 21 days, although in the first week it was realized more frequently (after 4 h, 1, 3, 5, and 7 days) by BLI and NIRF images (Figure 6). The BLI and NIRF signal were detectable only in the first week of evaluation. The BLI signal intensity increased over time until day 7 (Figure 6A, B), beginning with a signal approximately of 5 × 10^7^ photons/s in both groups after 4 h of cell implantation, achieving BLI values in the order of 10^9^ photons/s at 7-day. The “stroke + MSC + exercise” group showed a higher BLI signal intensity than “stroke + MSC” group at day 1 (2 × 10^8^ photons per second) until day 7 (7.4 × 10^9^ photons per second), as shown in Figure 6B. The NIRF signal showed an inverse behavior when compared to the BLI signal (Figure 6C), beginning with a high signal intensity, in the order of 10^8^ that seemed decreased over time (Figure 6C), with no difference between “stroke + MSC” and “stroke + MSC + exercise” being possible to observe.

## 4. Discussion

Local cellular therapy combined with physical exercise provided enhanced outcomes in reducing motor dysfunction, especially during the acute period, according to the findings of this study (7 days after stroke). When neurorehabilitation is the goal, this alternative strategy could be beneficial, especially if the pharmacological treatment window has closed.

In light of this, we must first investigate MSC tracking until brain lesion and attachment in cellular therapy. To do so, we employed MNP-labeled MSC and evaluated their internalization mechanism as well as their survivability after labeling. Fluorescence and brightfield microscopy pictures employing the TRITC and DAPI filters, as well as Prussian blue staining, which are extensively used for this evaluation [19,29], were used to confirm MNP internalization into MSC. MSC viability after 50 µg Fe per mL labeling was more than 98% in MTT assays and 97% in BLI assays, indicating higher rates than prior investigations of the group [13,17,30]. In a recent systematic review [13] of MSC tracking using contrast agents in a stroke model, physical–chemical characteristics of contrast agents were discussed as contributing to efficient internalization, as well as maintaining good cell viability and MSC ability to engraft, and thus allowing cell tracking by non-molecular imaging techniques. In a prior study [31], our lab demonstrated that MNP internalized into MSCs had no effect on cell viability or proliferation, preserving the cell’s ability to self-renew and differentiate. The intracellular location of MNPs into MSCs was also studied using transmission electron microscopy [31].

To induce small, well-defined infarctions suitable for comprehensive cell characterization or functional research, we used photothrombotic induction brain injury, which is routinely used in mouse stroke models, and able to generate motor impairment compatible with other models [32,33,34]. Due to its ability to replicate hemodynamic and inflammatory processes, photothrombotic models have a higher level of realism [32]. The ischemic brain lesion can be easily assessed and monitored in this model by a real-time reduction in blood perfusion, which is validated by TTC-unstained brain tissue, which indicates the infarcted area. A recent preclinical study [35] in rats that used the photothrombotic stroke model demonstrated a 60–78% loss in uniform blood perfusion in the damaged area. A similar effect was observed in this study and other research by the group [17,26], and it was closely linked to sensorimotor deficit [9]. 

MSC transplantation has recently been demonstrated to improve functional outcomes in experimental rodent models of localized ischemia [36], owing to cell processes such as neuronal replacement, promotion of angiogenesis, induction of brain plasticity, reduction of cell death, and immunomodulation [37]. The inflammatory response is considered to play a role in ischemic stroke pathogenesis [38], as well as recruiting stem cells to help with inflammation resolution and tissue repair [39]. MSCs have a significant immunomodulatory effect on both the adaptive and innate immune systems [40]. Paracrine mechanisms, such as secreted mediators (e.g., transforming growth factor, hepatocyte growth factor, prostaglandin E2), as well as metabolic activity (e.g., indoleamine 2,3-dioxygenas), appear to be involved in MSC-modulated lymphocyte suppression [41]. Finally, there is evidence that MSCs cause T-cells to polarize toward a regulatory phenotype, which may help to avoid inflammation [42], as well as provide trophic factors that help host neurons survive and regenerate [43]. Our findings revealed that transplanting MSC 24 h after ischemia improved the functional (motor) impairment in adult rats significantly. Other studies have found that in locus transplanted MSC improves functional outcomes after 7 days following a stroke [44,45]. 

Exercise may promote the therapeutic effect of mesenchymal cells by improving functional assessment as measured by behavioral tests, as well as inhibiting apoptosis in transplanted neuronal and MSC cells [12,46]. Furthermore, a recent systematic review [39] found that directing stem cells to regions of inflammation and tissue injury could explain the transitory increase in stem cell mobilization following exercise. The inflammatory process attracts stem cells to help with inflammation resolution and tissue repair, whereas exercise promotes cell mobilization by attracting extramedullary cells to areas of tissue injury and inflammation, such as skeletal muscle. Therefore, our results corroborate findings of a meta-analysis study [47] that found that post-ischemic exercise training starting between 1 to 5 days (mean exercise time of 14 days) was more effective in reducing infarct volume in the ischemic brain. Another important factor in the motor function improvement in the stroke model is the intensity of treadmill exercise modulation in the post-ischemic stage [48]. A high-intensity aerobic training was found to be more successful than a moderate-intensity aerobic training in improving aerobic fitness and regaining control of the impaired forelimb grip force, facilitating brain remodeling through a hemodynamic upgrade in the locus lesion [48].

The lack of a quantitative behavioral analysis associated with the functional molecular image of MSC in locus transplantation after exercise prompted us to investigate these parameters using the cylinder test and spontaneous motor activity test, which specified motor aspects of limb involvement, as well as global analysis of spontaneous movement, detailing coordination components for fast and slow movements, and horizontal and vertical, covered in the analyses of respective tests. The cylinder test revealed that after 7 days of brain damage, impairment of symmetry was identified, along with a relevant deficit of fast vertical spontaneous motor movement, similar to the findings of a previous study [26]. After 14 days, member symmetry improved with time, particularly in the group that received combination cell therapy and exercise, as well as increasing the frequency of fast vertical movements demanding more functional complexity of balance and coordination. The frequency of movement decreased over time in the other parameters, which can be explained by habituation to the test as the experimental groups approached the sham group behavioral pattern, primarily the group that received cells and physical training, followed by the groups that received only cells, and finally the group that received only exercise, as shown in Figure 5. Furthermore, this was the first study to link behavioral criteria to the presence of MSC in stroke lesion sites detected by BLI and FLI molecular imaging, allowing researchers to investigate the functional role of exercise-associated stem cells in rehabilitation.

In this regard, a meta-analysis [47] of different training strategies in animals submitted to ischemic stroke found that exercise training reduces functional impairment in animal stroke, analyzing 35 studies and more than 880 animals, particularly using the cylinder test performance, despite the fact the MSC adjuvant effect was not included in the meta-analysis evaluation. In contrast, a study that aimed to investigate the effect of MSC found that treadmill exercise enhances the therapeutic potency of transplanted bone mesenchymal stem cells in cerebral ischemic rats, with neurologic improvement 14 days after injury [12]. As a result of our findings, we were able to confirm that cell treatment and exercise resulted in long-term functional recovery. In addition, a study [49] that used stem cells treatment until 1 or 3 days after stroke revealed long-term functional recovery in young and old mice of both sexes, as well as non-human primates (marmosets). Mice that were given MSC had the deficit for 14 days and had a lower preference for using their unaffected (left) forelimb over their damaged (right) forelimb. The cylinder test was used to evaluate the behavior for 8 weeks, and the same behavior evolution was found in marmosets using the neuroscale, confirming our findings. 

Therefore, these spontaneous limb motion tests (cylinder test) and spontaneous movement on the ground test can provide useful information about locomotor impairment (initial and chronic limb deficiencies) after stroke induction and their recovery over time evaluation, particularly when quantitative and qualitative parameters are examined [16,50], as demonstrated in our study. 

In terms of molecular imaging, previous studies have employed techniques including BLI [51], NIRF [52], and MRI [53] to track MSC in stroke models. Since each molecular imaging technique has its own advantages and limitations, these methods combined allowed us to optimize the MSC follow-up while taking into account the inherent limitations of each technique. The combination of BLI and NIRF approaches allowed the assessment particle internalization as a result of MSC luciferase expression and MNP attachment to a fluorescent agent. BLI has the advantage of maintaining luciferase protein expression even after cell division, which nanoparticles do not [54]. BLI and NIRF techniques were used to track transplanted MSC for up to 7 days in our study, with the latter demonstrating an inverse signal relationship. In both groups (the one that received only cells and the other that received cells and exercise), BLI showed a growing signal intensity that started in the order of 5 × 10^7^ photons per second, reaching values of 10^9^ photons per second in 7 days. When compared to the group that simply received cells, the group that received cells and exercise exhibited a stronger signal over time. The NIRF signal began with intensity in the order of 10^8^ and gradually declined over time, with no discernible difference across groups. There was no signal detectable for any of these techniques in the 14-day analysis. BLI allows tumors and other cells to be monitored for lengthy periods of time [19,55]. MSC could not be tracked for more than one week in a stroke model in previous studies from our group [17]. This brief period of evaluation could be attributed to the activation of the innate immune system, which is responsible for clearance of transplanted cells and can be stimulated by MSC after implantation [56]. As a result, the animal model chosen appears to have a significant impact on the outcome of cell therapy [57]. 

Along with homing, the release of physiologically active chemicals such as cytokines, growth factors, and chemokines known as the MSC secretome may be the key to successful cell therapy. The action of phosphatidylinositol-3-kinase (phosphoinositide 3-kinase) is one of the intracellular signaling mechanism (PI3K) Akt (serine-threonine protein kinase) pathway to protein kinase B (serine-threonine protein kinase Akt) pathways [58]. This study did not directly analyze the PI3K/Akt pathway, which plays a role in the survival of MSCs [58], but this analysis was ensured by noninvasive molecular image, showing MSCs survival and proliferation by the increased intensity signal of the bioluminescence image over time in the brain injury region, accompanying a functional improvement.

There are certain limitations to our study. Physical activity was done after the stroke was induced, but in the future, physical training may be done before the stroke to see if it would improve the cell treatment effectiveness. Variations in the intensity of physical activity that could have altered the therapeutic process were not investigated in the same physical training. The study used molecular imaging to track cells and the animal’s functional progression after a stroke, as well as novel and combination therapy techniques. In the future, however, we will concentrate on structural analysis using immunohistochemistry and histology.

## 5. Conclusions

The goal of our research was to see how the combination of MSC and exercise affected the motor recovery of rats who had suffered an ischemic stroke. The group that received both cells and exercise was able to do more complex movements than the groups that just received one intervention (cells or physical training), and had faster recovery of symmetry over time. The cell viability in the wounded brain area was assessed using the BLI technique up to 7 days after the injury, and the group that received cells and exercise showed higher signal intensity over time, indicating more favorable conditions for cell proliferation. These data suggest that combining workouts with the use of MSC for the treatment of ischemic stroke could be beneficial.

## Figures and Tables

**Figure 1 cells-11-00485-f001:**
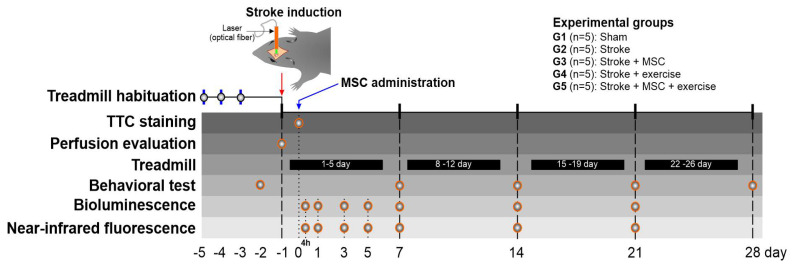
The experimental design included the temporal distribution of pre- and post-stroke induction evaluations, as well as the experimental groups (G1–G5). Treadmill habituation at −4, −3, −2 days before stroke induction, TTC staining at 1 day after stroke induction, perfusion evaluation pre- and post-stroke induction, treadmill physical training for 4 weeks (5 days per week), behavioral testing at −1 (control), and 7, 14, 21, and 28 days. At 4 h (28 h after stroke induction), and 1, 3, 5, 7, 14, and 21 days following MSC delivery, bioluminescence and near-infrared fluorescence were observed. Abbreviations: TTC (2,3,5-triphenyltetrazolium chloride staining); MSC (mesenchymal stem cells).

**Figure 2 cells-11-00485-f002:**
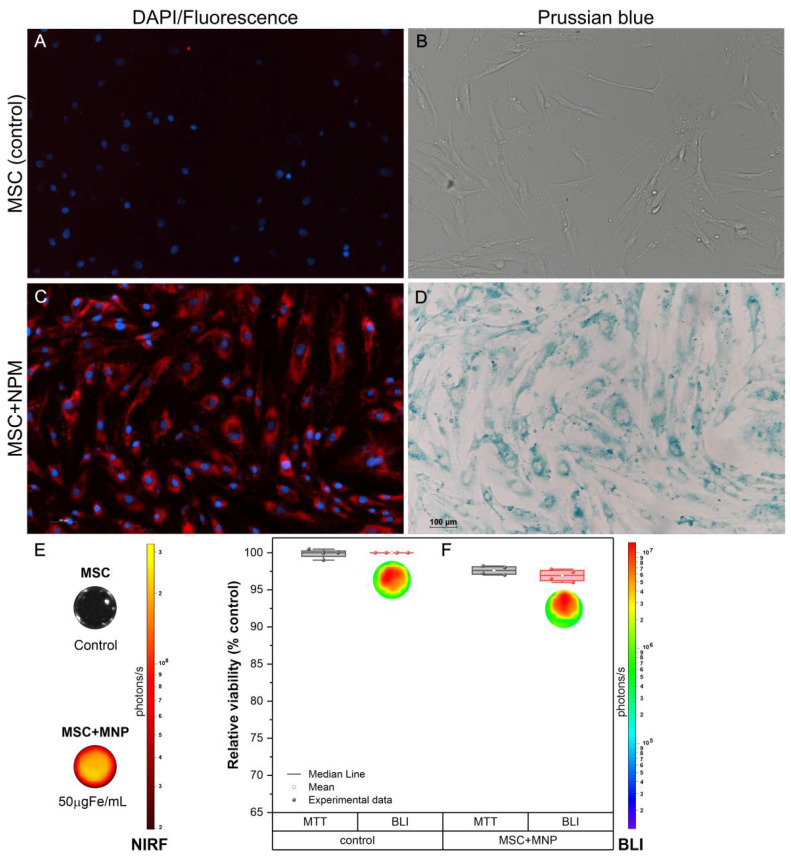
Internalization and viability of fluorescence–magnetic nanoparticles labeled mesenchymal stem cells. (**A**) Fluorescence microscopy image of unlabeled MSC at 10×, (**B**) Brightfield microscopy image of unlabeled MSC at 10×, (**C**) TRITC and DAPI fluorescence microscopy images of MSC labeled with 50 µg Fe/mL of MNP, displayed at 10×, (**D**) Brightfield microscopy image of MSC labeled with 50 µg Fe/mL of NP, shown at 10×, (**E**) NIRF signal of MSC labeled with 50 µg Fe per mL of MNP, and (**F**) Cellular viability of MSC following labeling in MTT (gray bars) and BLI (red bars) assays, respectively. A red–yellow scale bar represented the NIRF intensity in photons per second, and the BLI intensity in photons per second was represented by blue–red scale bar. Abbreviations—MSC: mesenchymal stem cells from the human bone marrow; MNP: magnetic nanoparticles; MTT: the 3-[4,5-dimethylthiazol-2-yl]-2,5 diphenyl tetrazolium bromide; BLI: bioluminescent images.

**Figure 3 cells-11-00485-f003:**
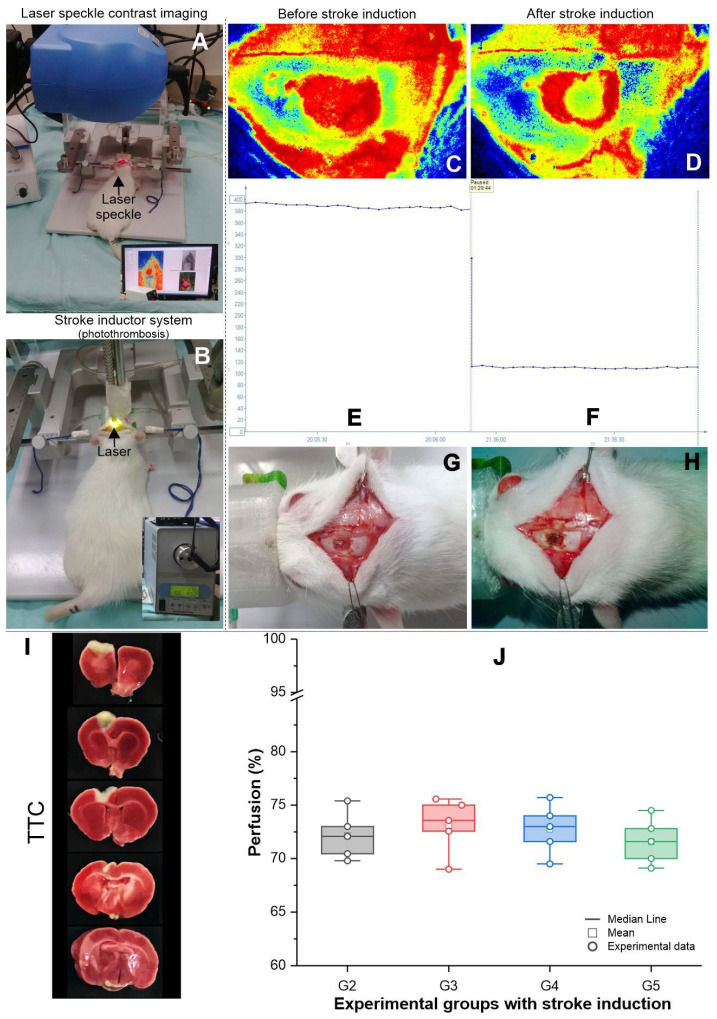
Evaluation of focal ischemic brain lesion by photothrombosis. (**A**) The laser speckle imagining system used for local blood perfusion evaluation; (**B**) The laser equipment used induce focal ischemic lesion by photothrombosis; (**C**) Baseline local blood perfusion before stroke induction; (**D**) Local perfusion after stroke induction; (**E**,**F**) The graphic representation of blood perfusion before and after stroke induction respectively; (**G**,**H**) Visualization of the animal’s skull with craniectomy before and after stroke induction, respectively; (**I**) Evaluation of stroke induction by TTC; and (**J**) Box plot of local blood perfusion change in the G2–G5 experimental groups. G2: Stroke group; G3: Stroke + MSC group; G4: Stroke + exercise group; G5: Stroke + MSC + exercise group; SD: Standard deviation.

**Figure 4 cells-11-00485-f004:**
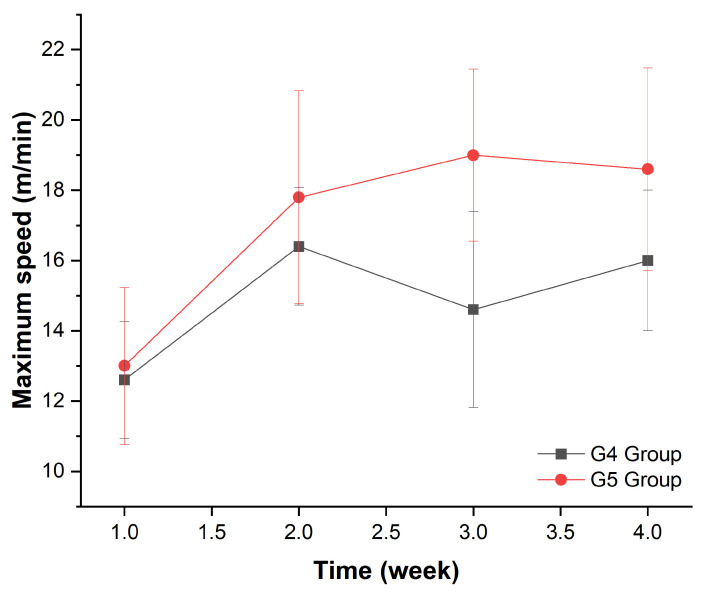
Maximum speed evolution during four weeks of treadmill training in the G4 and G5 groups.

**Figure 5 cells-11-00485-f005:**
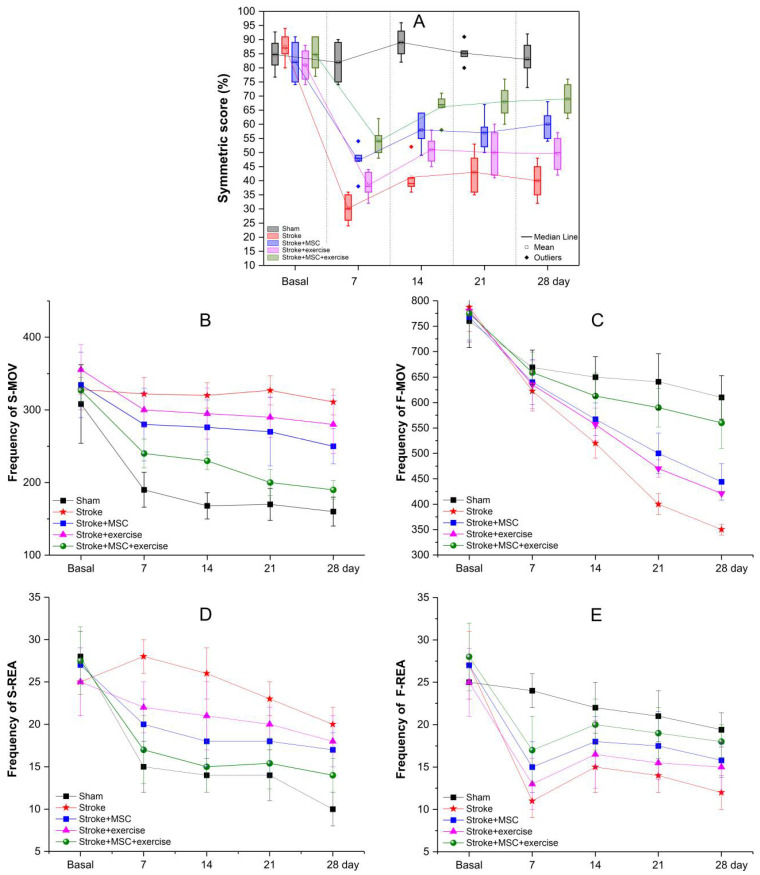
Behavioral evaluation by the cylinder test and the spontaneous locomotor activity test before stroke induction and at 7, 14, 21, and 28 days after in experimental groups (represented by the red star, blue square, pink triangle, and green circle) and control group (black square symbol). (**A**) over time by the cylinder test; (**B**–**E**) Motor evaluation by the spontaneous locomotor activity test considering the frequency of the following movements—(**B**) Slow horizontal movements (S-MOV); (**C**) Fast horizontal movements (F-MOV); (**D**) Slow vertical movements (S-REA); and (**E**) Fast vertical movements (F-REA).

**Figure 6 cells-11-00485-f006:**
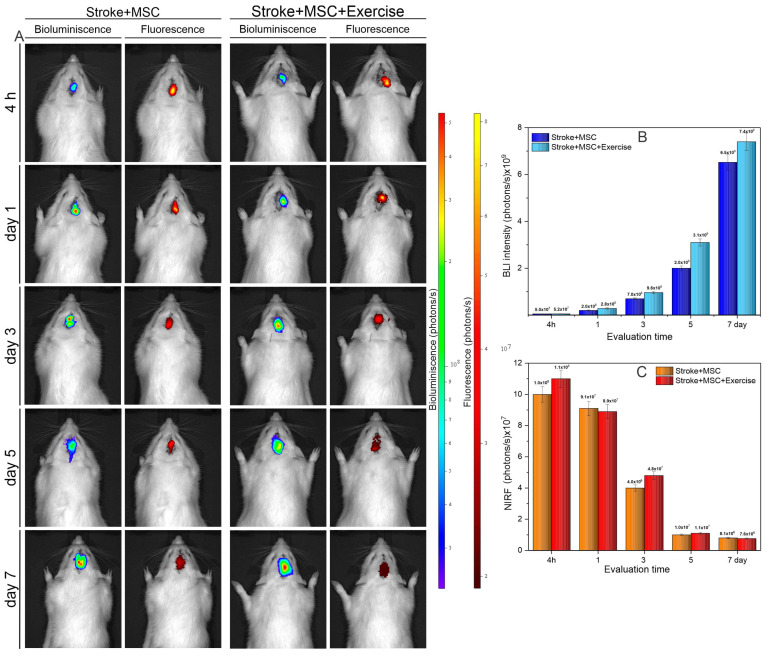
Evaluation of bioluminescence and fluorescence signal in the “stroke + MSC” (G3) and “stroke + MSC + exercise” (G5) groups, in the following conditions: 4 h and 1, 3, 5, 7, 14, and 21 days after the administration of MSC labeled with MNP. (**A**) BLI and NIRF signal representations in the “stroke + MSC” (G3) and “stroke + MSC + exercise” (G5) groups over the time with detectable signal for both techniques (until 7 day); (**B**) The graphic of the BLI signal intensity of groups analyzed over time; and (**C**) The graphic of the NIRF signal intensity of groups analyzed over time.

**Table 1 cells-11-00485-t001:** MSC cellular viability in MTT and BLI assays following labeling.

[Fe] (µg Fe/mL)	MTT (%) Mean ± SD	BLI (%) Mean ± SD	*n*
[0]	99.875 ± 0.629	100	4
[50]	97.600 ± 0.589	96.925 ± 0.822	4

Abbreviations—MTT: the 3-[4,5-dimethylthiazol-2-yl]-2,5 diphenyl tetrazolium bromide; BLI: bioluminescence imaging; SD: standard deviation.

**Table 2 cells-11-00485-t002:** The local blood perfusion difference, comparing the baseline perfusion after ischemic induction perfusion in each group submitted to stroke model (G2, G3, G4, and G5).

Group	Blood Perfusion Mean ± SD	*n*
G2	72.15 ± 2.22	5
G3	73.14 ± 2.60	5
G4	72.76 ± 2.36	5
G5	71.60 ± 2.16	5

Abbreviations—G2: stroke group; G3: stroke + MSC group; G4: stroke + exercise group; G5: stroke + MSC + exercise group; SD: standard deviation.

**Table 3 cells-11-00485-t003:** ANOVA of the local blood perfusion difference, compared with a baseline after ischemic induction perfusion in each group submitted to stroke model (G2, G3, G4, and G5).

ANOVA Test					
Cases	Sum of Squares	df	Mean Square	F	*p*
Groups	6.88	3	2.29	0.419	0.742
Residuals	87.54	16	5.47		

Note. Type III sum of squares.

**Table 4 cells-11-00485-t004:** The maximum speed evaluation over 4 weeks of treadmill training in G4 and G5 groups.

Week	Maximum Speed Mean ± SD (m Per Min)		N
	G4	G5	
1	12.60 ± 1.67	13.00 ± 2.24	10
2	16.40 ± 1.67	17.80 ± 3.03	10
3	14.60 ± 2.79	19.00 ± 2.45	10
4	16.00 ± 2.00	18.60 ± 2.88	10

**Table 5 cells-11-00485-t005:** ANOVA test of the maximum speed evaluated during the course of four weeks of treadmill training in the G4 and G5 groups.

ANOVA Test					
Cases	Sum of Squares	df	Mean Square	F	*p*
Weeks	137.800	3	45.933	8.006	<0.001
Groups	48.400	1	48.400	8.436	0.007
Week * Group	22.200	3	7.400	1.290	0.295
Residuals	183.600	32	5.737		

Note: * interaction analysis between the week and group factors.

## Data Availability

Data available via request.

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
