# Peer review of "Effect of Cell Therapy and Exercise Training in a Stroke Model, Considering the Cell Track by Molecular Image and Behavioral Analysis"

_cells, 2022, doi:10.3390/cells11030485_

Round 1

Reviewer 1 Report

Authors should address the below notes before publication of the manuscript:

  1. please add more points in conclusion and discussion.
  2. Please elaborate the reason for choosing the MSCs with bone marrow origin.

Author Response

Reviewer #1

Authors should address the below notes before publication of the manuscript:

  1. Please add more points in conclusion and discussion.

Answer:  Thank you for your time and dedication to the manuscript review. We added more information to improve the explanation about the finds in the discussion and conclusion sections of the manuscript.

  1. Please elaborate the reason for choosing the MSCs with bone marrow origin.

Answer:  Thank you for suggestion. We described the choice of MSCs from bone marrow in the introduction and discussion sections of the manuscript.

Reviewer 2 Report

The submitted manuscript contains a high amount of analyzed data with different tools and information about MSC+exercise on stroke, adding a piece of information in the comprehension of MSC and exercise.

  1. The manuscript contains several typographical and grammatical errors that need to be corrected.
  2. As the proliferation in the injury site increases, the molecular mechanism needed to be explored such as PAKT pathway.
  3. As the recovery of stroke is a complex event including more than proliferation, we suggested that the author should evaluate the recovery of stroke in more aspects (in molecular mechanism)

Author Response

Reviewer #2

The submitted manuscript contains a high amount of analyzed data with different tools and information about MSC+exercise on stroke, adding a piece of information in the comprehension of MSC and exercise.

  1. The manuscript contains several typographical and grammatical errors that need to be corrected.

Answer:  Thank you for your time and dedication to the manuscript review. We corrected all typographical and grammatical errors in this new version of the manuscript.  

  1. As the proliferation in the injury site increases, the molecular mechanism needed to be explored such as PAKT pathway.

Answer:  Thank you for your suggestion. We added more explanations about the PAKT pathway in the discussion section of the manuscript. Along with homing, the release of physiologically active chemicals such as cytokines, growth factors, and chemokines known as the MSC secretome may be the key to successful cell therapy. The action of phosphatidylinositol-3-kinase (phosphoinositide 3-kinase) is one of the intracellular signaling mechanisms (PI3K) Akt (serine-threonine protein kinase) pathway - protein kinase B (serine-threonine protein kinase Akt) pathway [1]. The PI3K/Akt pathway has a role in the survival of MSCs, evidenced by the effects of overexpression of key components of this pathway [2]. Activation of Akt plays a crucial role in basic cellular functions such as cell proliferation and survival [3]. This study did not directly analyze the PAKT pathway, but the MSC survival and proliferation detectable by their increase of bioluminescence intensity signal over time and the functional improvement indirectly evidenced the efficiency of MSCs and the activity of this important molecular mechanism of stroke recovery.

  1. As the recovery of stroke is a complex event including more than proliferation, we suggested that the author should evaluate the recovery of stroke in more aspects (in molecular mechanism)

Answer: Thank you for your suggestion, but the main goal of the study was to evaluate cellular therapy with mesenchymal stem cells following stroke induction with or without exercise using a behavioral test and non-invasive molecular imaging to track MSC localization (cell homing and tracking) during recovery. As a result, by ensuring cell homing and viability at the site of damage in this study, we may be more confident in the MSCs molecular pathways in ischemia recovery, which have been established in other studies [4-6] and contribute to functional recovery, as shown in our results.

This limitation of our study was described in the last paragraph of the manuscript discussion, and will be carried out in future studies.

References

  1. Samakova, A.; Gazova, A.; Sabova, N.; Valaskova, S.; Jurikova, M.; Kyselovic, J. The PI3k/Akt pathway is associated with angiogenesis, oxidative stress and survival of mesenchymal stem cells in pathophysiologic condition in ischemia. Physiological research 2019, 68, S131-s138, doi:10.33549/physiolres.934345.
  2. Samakova, A.; Ga~ová, A.; Sabova, N.; Valáaková, S.; Jurikova, M.; Kyselovj , J. The PI3k/Akt pathway is associated with angiogenesis, oxidative stress and survival of mesenchymal stem cells in pathophysiologic condition in ischemia. Physiological research 2019, 68 Suppl 2, S131-S138.
  3. Wang, X.L.; Qi, J.; Shi, Y.Q.; Lu, Z.Y.; Li, R.L.; Huang, G.J.; Ning, B.B.; Hao, L.S.; Wang, H.; Hao, C.N., et al. Atorvastatin plus therapeutic ultrasound improve postnatal neovascularization in response to hindlimb ischemia via the PI3K-Akt pathway. American journal of translational research 2019, 11, 2877-2886.
  4. Li, Y.; Dong, Y.; Ran, Y.; Zhang, Y.; Wu, B.; Xie, J.; Cao, Y.; Mo, M.; Li, S.; Deng, H., et al. Three-dimensional cultured mesenchymal stem cells enhance repair of ischemic stroke through inhibition of microglia. Stem Cell Research & Therapy 2021, 12, 358, doi:10.1186/s13287-021-02416-4.
  5. Gu, Y.; Zhang, Y.; Bi, Y.; Liu, J.; Tan, B.; Gong, M.; Li, T.; Chen, J. Mesenchymal stem cells suppress neuronal apoptosis and decrease IL-10 release via the TLR2/NFκB pathway in rats with hypoxic-ischemic brain damage. Molecular Brain 2015, 8, 65, doi:10.1186/s13041-015-0157-3.
  6. Do, P.T.; Wu, C.-C.; Chiang, Y.-H.; Hu, C.-J.; Chen, K.-Y. Mesenchymal Stem/Stromal Cell Therapy in Blood–Brain Barrier Preservation Following Ischemia: Molecular Mechanisms and Prospects. International Journal of Molecular Sciences 2021, 22, 10045.

Reviewer 3 Report

Nucci et al., have performed a study to check the effect of cell therapy and exercise on motor-behavioral outcomes following stroke. The study is robust, the data look convincing and convey the message clearly. However, few questions are to be addressed.

  1. Did the properties of MSCs change following internalization of MNPs by MSCs? Did the authors perform other studies apart from MMT assay to check the viability? Also, a 3D fluorescence imaging would be useful in actually detect the internalized MNPs by the MSCs as a 2D microscopy images will not be able to distinguish properly between the MNPs actually internalized by the MSCs from those that might be on the surface of the MSCs and not removed by PBS washings.
  2. The infarct developed by the model is very small as compared to infarct developed by other clinically relevant models such as MCAo. The defects developed from the model used would not have been very significant looking from the size of the infarct seen from the TTC stain. How will the authors address the same?
  3. Not much has been discussed related to the mechanism behind this improvement. What factors are involved in the motor-behavioural improvements seen over time in animals? What role does MSCs play in the same? Discussion about the same is essential.

Author Response

Reviewer #3

Nucci et al., have performed a study to check the effect of cell therapy and exercise on motor-behavioral outcomes following stroke. The study is robust, the data look convincing and convey the message clearly. However, few questions are to be addressed.

  1. Did the properties of MSCs change following internalization of MNPs by MSCs? Did the authors perform other studies apart from MMT assay to check the viability? Also, a 3D fluorescence imaging would be useful in actually detect the internalized MNPs by the MSCs as a 2D microscopy images will not be able to distinguish properly between the MNPs actually internalized by the MSCs from those that might be on the surface of the MSCs and not removed by PBS washings.

Answer:  Thank you for your time and dedication to the manuscript review. Our group showed in a previous study [1] that MMPs internalized into MSCs did not influence their viability or proliferation, maintaining the cell capacity in self-renewal and differentiation. Regarding cell viability, we also analyzed by bioluminescence image due to the ability of these MSCs to express luciferase after lentiviral transduction.

Thank you for your suggestion regarding the 3D fluorescence image, but we do not have the equipment available to perform this type of analysis, however, in previous studies [1] carried out in our group, the intracellular localization of MNP in MSCs was shown through the transmission electron microscopy technique. This information was included in the discussion section of the manuscript.

  1. The infarct developed by the model is very small as compared to infarct developed by other clinically relevant models such as MCAo. The defects developed from the model used would not have been very significant looking from the size of the infarct seen from the TTC stain. How will the authors address the same?

Answer:  Thank you for your observation. The photothrombotic stroke preclinical model is reproducible, relatively non-invasive, and technically simple [2,3]. Two cortical areas in rodents are involved in the control of skilled forelimb movements. The caudal forelimb area (CFA) is the primate equivalent of the primary motor cortex (M1), and its caudal border area includes the primary somatosensory forelimb area. The rostral forelimb area (RFA) is a second motor area that is more rostrally located and has been regarded as a premotor area due to similarities in connectivity and neuronal responses associated with movement and pre-movement planning [4]. A systematic review performed by our group [5] showed that the infarct area by photothrombotic is smaller than the MCAo stroke model, but both models showed behavior impairment compatible with the CFA and RFA motor brain regions commitment, as also shown in this study through the forelimb movement impairments (spontaneous movement and symmetry) analyzed by behavioral tests.

  1. Not much has been discussed related to the mechanism behind this improvement. What factors are involved in the motor-behavioural improvements seen over time in animals? What role does MSCs play in the same? Discussion about the same is essential.

Answer:  Thank you for your time and dedication in the manuscript review. The inflammation process is considered a key contributor to the pathophysiology of ischemic stroke and is strongly associated with infarct size[6], and this process attracts stem cells to help with inflammation resolution and tissue repair [7]. The key aim of the study was analysis cellular therapy with mesenchymal stem cells after stroke induction with or without exercise by behavioral test, monitoring the MSC localization (cells homing and tracking) during recovery by non-invasive molecular imaging. In other papers of the group, the infarct size and other molecular aspects were stronger associated with effective recovery. MSCs have strong immunomodulatory activity on the adaptive and innate immune systems [8]. The MSC-modulated lymphocyte suppression seems to be due to paracrine mechanisms, including secreted mediators (eg, transforming growth factor-β, hepatocyte growth factor, prostaglandin E2), as well as metabolic activity (eg, indoleamine 2,3-dioxygenas) [9]. Finally, there is substantial evidence that MSCs induce polarization of T cells toward a regulatory phenotype, which may also prevent inflammation [10], as well as produce trophic factors that induce survival and regeneration of host neurons [11].

References

  1. Sibov, T.T.; Pavon, L.F.; Miyaki, L.A.; Mamani, J.B.; Nucci, L.P.; Alvarim, L.T.; Silveira, P.H.; Marti, L.C.; Gamarra, L. Umbilical cord mesenchymal stem cells labeled with multimodal iron oxide nanoparticles with fluorescent and magnetic properties: application for in vivo cell tracking. International journal of nanomedicine 2014, 9, 337-350, doi:10.2147/IJN.S53299.
  2. Labat-gest, V.; Tomasi, S. Photothrombotic ischemia: a minimally invasive and reproducible photochemical cortical lesion model for mouse stroke studies. J Vis Exp 2013, 10.3791/50370, doi:10.3791/50370.
  3. Jolkkonen, J.; Jokivarsi, K.; Laitinen, T.; Gröhn, O. Subacute hemorrhage and resolution of edema in Rose Bengal stroke model in rats coincides with improved sensorimotor functions. Neurosci Lett 2007, 428, 99-102, doi:10.1016/j.neulet.2007.09.043.
  4. Neafsey, E.J.; Sievert, C. A second forelimb motor area exists in rat frontal cortex. Brain Res 1982, 232, 151-156, doi:10.1016/0006-8993(82)90617-5.
  5. Nucci, L.P.; Silva, H.R.; Giampaoli, V.; Mamani, J.B.; Nucci, M.P.; Gamarra, L.F. Stem cells labeled with superparamagnetic iron oxide nanoparticles in a preclinical model of cerebral ischemia: a systematic review with meta-analysis. Stem Cell Res Ther 2015, 6, 27, doi:10.1186/s13287-015-0015-3.
  6. Xiong, X.Y.; Liu, L.; Yang, Q.W. Functions and mechanisms of microglia/macrophages in neuroinflammation and neurogenesis after stroke. Progress in neurobiology 2016, 142, 23-44, doi:10.1016/j.pneurobio.2016.05.001.
  7. Emmons, R.; Niemiro, G.M.; De Lisio, M. Exercise as an Adjuvant Therapy for Hematopoietic Stem Cell Mobilization. Stem Cells Int 2016, 2016, 7131359, doi:10.1155/2016/7131359.
  8. Le Blanc, K.; Mougiakakos, D. Multipotent mesenchymal stromal cells and the innate immune system. Nature reviews. Immunology 2012, 12, 383-396, doi:10.1038/nri3209.
  9. Nauta, A.J.; Fibbe, W.E. Immunomodulatory properties of mesenchymal stromal cells. Blood 2007, 110, 3499-3506, doi:10.1182/blood-2007-02-069716.
  10. English, K. Mechanisms of mesenchymal stromal cell immunomodulation. Immunology and cell biology 2013, 91, 19-26, doi:10.1038/icb.2012.56.
  11. Crigler, L.; Robey, R.C.; Asawachaicharn, A.; Gaupp, D.; Phinney, D.G. Human mesenchymal stem cell subpopulations express a variety of neuro-regulatory molecules and promote neuronal cell survival and neuritogenesis. Exp Neurol 2006, 198, 54-64, doi:10.1016/j.expneurol.2005.10.029.

Round 2

Reviewer 3 Report

The comments are addressed. The manuscript may be considered for publication.